# Regulation of EZH2 Expression by INPP4B in Normal Prostate and Primary Prostate Cancer

**DOI:** 10.3390/cancers15225418

**Published:** 2023-11-15

**Authors:** Manqi Zhang, Yasemin Ceyhan, Shenglin Mei, Taghreed Hirz, David B. Sykes, Irina U. Agoulnik

**Affiliations:** 1Division of Medical Oncology, Department of Medicine, Duke University, Durham, NC 27708, USA; manqi.zhang@duke.edu; 2Public Health Sciences Division, Fred Hutchinson Cancer Center, Seattle, WA 98109, USA; yceyhan@fredhutch.org; 3Center for Regenerative Medicine, Massachusetts General Hospital, Boston, MA 02114, USA; smei8@mgh.harvard.edu (S.M.); thirz@mgh.harvard.edu (T.H.); dbsykes@mgh.harvard.edu (D.B.S.); 4Harvard Stem Cell Institute, Cambridge, MA 02138, USA; 5Department of Human and Molecular Genetics, Herbert Wertheim College of Medicine, Florida International University, Miami, FL 33199, USA; 6Biomolecular Science Institute, Florida International University, Miami, FL 33199, USA; 7Department of Molecular and Cellular Biology, Baylor College of Medicine, Houston, TX 77030, USA

**Keywords:** INPP4B, PTEN, EZH2, prostate cancer

## Abstract

**Simple Summary:**

Prostate cancer is a heterogeneous disease driven by multiple genetic alterations: the deletion or downregulation of tumor suppressors and the activation or amplification of oncogenes. Among the most frequently deleted tumor suppressors in prostate cancer are INPP4B and PTEN. We show that the loss of these proteins triggers distinct compensatory mechanisms that must be overcome for the progression from indolent to advanced stages of prostate cancer.

**Abstract:**

The phosphatases INPP4B and PTEN are tumor suppressors that are lost in nearly half of advanced metastatic cancers. The loss of PTEN in prostate epithelium initially leads to an upregulation of several tumor suppressors that slow the progression of prostate cancer in mouse models. We tested whether the loss of INPP4B elicits a similar compensatory response in prostate tissue and whether this response is distinct from the one caused by the loss of PTEN. Knockdown of INPP4B but not PTEN in human prostate cancer cell lines caused a decrease in EZH2 expression. In *Inpp4b^−/−^* mouse prostate epithelium, EZH2 levels were decreased, as were methylation levels of histone H3. In contrast, *Ezh2* levels were increased in the prostates of *Pten^−/−^* male mice. Contrary to PTEN, there was a positive correlation between INPP4B and EZH2 expression in normal human prostates and early-stage prostate tumors. Analysis of single-cell transcriptomic data demonstrated that a subset of EZH2-positive cells expresses INPP4B or PTEN, but rarely both, consistent with their opposing correlation with EZH2 expression. Unlike PTEN, INPP4B did not affect the levels of SMAD4 protein expression or *Pml* mRNA expression. Like PTEN, p53 protein expression and phosphorylation of Akt in *Inpp4b^−/−^* murine prostates were elevated. Taken together, the loss of INPP4B in the prostate leads to overlapping and distinct changes in tumor suppressor and oncogenic downstream signaling.

## 1. Introduction

Inositol polyphosphate-4-phosphatase type II B (INPP4B) and phosphatase and tensin homolog (PTEN) are dual-specificity phosphatases that play significant roles in phosphatidylinositol signaling. INPP4B preferentially binds and dephosphorylates PI(3,4)P2 and PI(4,5)P2, leading to inhibition of the Akt and PKC pathways, respectively [1,2,3]. PTEN dephosphorylates PI(3,4,5)P3, suppressing Akt signaling. Both INPP4B and PTEN are tumor suppressors in prostate cancer [2]. A previous study showed that INPP4B expression is decreased in 8% of clinically localized diseases and 47% of metastatic samples [4]. We have previously shown that in rapidly proliferating tumors represented by high Ki67 expression, the loss of INPP4B coincides with accelerated recurrence [2]. Both INPP4B and PTEN display modest rates of copy number alterations (CNAs) in primary tumors and a significant loss of expression in metastatic cancers, with the corresponding increase in PI3K/Akt signaling in all metastases [4].

The loss of PTEN during the early stages of prostate cancer elicits compensatory changes that oppose tumorigenesis, and additional mutations are required to stimulate progression to advanced stages of prostate cancer [5,6]. In mice, *Pten* loss (prostate-specific) leads to the development of indolent tumors with long latency and a minimally invasive phenotype [5]. The long latency is the result of activation of the tumor-suppressing TRP53, SMAD4, and PML pathways, which oppose progression and metastases in cells with PTEN loss [7,8,9,10,11]. Genes encoding these tumor suppressors are also frequently mutated in human prostate cancer. Consistent with this finding, in mice with prostate-specific *Pten* knockout, concomitant deletion of any one of these genes leads to the development of aggressive metastatic prostate tumors [7,8,11].

Prostate cancer initiation and progression are dependent on androgen receptor (AR) signaling, which changes from promoting differentiation to stimulating proliferation and metastasis [12,13]. In addition to increased AR protein levels and transcriptional output, significant reprogramming of the AR cistrome and transcriptome accompanies prostate cancer progression. The epigenetic reprogramming and changes in expression are due, in part, to the enhancer of zeste homolog 2 (EZH2), an oncogene that is overexpressed in various cancers including prostate, breast, bladder, endometrial, and small-cell lung cancers [14,15]. In prostate cancer, EZH2 functions both as an epigenetic writer and AR coregulator. EZH2 is the catalytic subunit of polycomb repressive complex 2 (PRC2), a histone-lysine methyltransferase, which induces epigenetic reprogramming by tri-methylation of lysine 27 (K27) and di-methylation of lysine 9 (K9) on histone H3 [16,17,18]. The EZH2 coactivator function is PRC2-independent and promotes ligand-independent AR activity, mediating the development of castration-resistant prostate cancer (CRPC) [14,19]. A shift from a polycomb repressor to an AR transcriptional coregulator is mediated by the phosphorylation of EZH2 on S21 by Akt [20], and EZH2 S21 phosphorylation stimulates cellular proliferation and accelerates tumor growth in mouse models [21]. Clinical trials are ongoing to investigate EZH2 inhibition as a treatment for CRPC [NCT03480646] [15,22]. Along with EZH2 reprogramming, overactive PI3K/Akt signaling in prostate cancer was shown to cause mTORC1-dependent upregulation of *TP53* translation [4,23].

We have previously reported that INPP4B is a tumor suppressor in human prostate cancer [2]. Similar to *Pten^loxP/loxP^*; *PB4-Cre*, the deletion of *Inpp4b* alone is insufficient for the development of invasive prostate cancer, despite increased phosphorylation levels of Akt [5,24] and increased prostate inflammation [25]. We investigated whether this might be due to the activation of compensatory mechanisms triggered by the loss of INPP4B that opposes tumor progression.

In this report, we compare the changes that result from the loss of INPP4B to those caused by the loss of PTEN. We describe common and distinct compensatory changes caused by the loss of INPP4B in human cell lines, mouse prostate, and normal human prostate epithelium and prostate cancers. Knockdown of INPP4B reduces the EZH2 in prostate cancer cell lines on both the RNA and protein levels. In vivo, the prostates of *Inpp4b^−/−^* mice express reduced EZH2 protein levels compared to WT mice. Consistent with decreased prostatic levels in EZH2, the substrates of its methyl transferase activity, H3K27me3 and H3K9me2, are also decreased. We observe a positive correlation between *INPP4B* and *EZH2* expression in the human prostate epithelium and primary adenocarcinomas and, consistent with previous reports, a negative correlation between PTEN and EZH2. Similar to PTEN, protein levels of p53 are significantly increased in the prostates of *Inpp4b^−/−^* mice. Unlike PTEN-null prostate, the expression of *Pml* and protein levels of SMAD4 remain unchanged in *Inpp4b^−/−^* males. Increased levels of PTEN and downregulation of EZH2 might contribute to the indolent nature of the prostate phenotype in *Inpp4b^−/−^* males and in the prostate epithelium and early prostate cancer in men.

## 2. Materials and Methods

### 2.1. Cell Culture Reagents and Compounds

The prostate cancer cell lines LNCaP and VcaP were purchased from ATCC and were maintained according to the manufacturer’s recommendation. All the media were purchased from Thermo Fisher Scientific (Waltham, MA, USA); fetal bovine serum (FBS) and charcoal-stripped serum (CSS) were purchased from Sigma-Aldrich (St. Louis, MO, USA). Enzalutamide and bicalutamide were purchased from Selleckchem (S1250 and S1190, Houston, TX, USA). The CSS treatments were performed in a consecutive manner.

### 2.2. Reverse Phase Protein Array (RPPA)

LNCaP cells were incubated in a medium supplemented with 10% FBS and were transfected with control siRNA or INPP4B siRNA for 48 h. Four biological replicates were used for the control and INPP4B knockdown groups. The proteins were extracted using a tissue protein extraction reagent (TPER, Pierce) supplemented with 450 mM NaCl and a cocktail of protease and phosphatase inhibitors (Roche Life Science, Basel, Switzerland). The RPPA was conducted at the Cancer Proteomics and Metabolomics Core Facility (Baylor College of Medicine, Houston, TX, USA), as previously described [26,27]. Briefly, each sample was arrayed in triplicate on nitrocellulose-coated slides using an Aushon 2470 Arrayer. Immunostaining was performed on an automated slide stainer, Autolink 48 (Dako, Glostrup, Denmark). The Catalyzed Signal Amplification System kit (Dako) and fluorescent IRDye 680 Streptavidin (LI-COR) were used as the detection system. The slides were scanned by a GenePix Axon AL4200 Scanner (Molecular Probes, San Jose, CA, USA). Total protein values were assessed by staining one or more slides with Sypro Ruby Blot Stain (Molecular Probes). The data were normalized using a group-based normalization method. The distribution of the normalized data was summarized into mean, SD, range, median, and quartiles and tested for normality using Shapiro–Wilk and Shapiro–Francia tests. One-way analysis of variance (ANOVA) was used to analyze differences in mean expression levels among the groups, and Duncan’s multiple comparison procedure was used to determine which means differed. The family-wise error rate was controlled using the Bonferroni method at a 0.05 level.

### 2.3. Gene Set Enrichment Analysis (GSEA)

The GSEA was performed as previously described [28]. The gene set of control and INPP4B knockdown LNCaP cells were previously generated in our lab (GSE111725) [28]. The EZH2 signatures (Appendix A) were acquired from data set GSE39452, which includes 266 genes significantly changed (*p* < 0.01) with knockdown of EZH2 [20], and data set GSE107779, which includes 824 genes with significant changes for more than 2 folds [29].

### 2.4. siRNA Transfection

The LNCaP or VCaP cells were transfected with either negative control or INPP4B siRNAs for 48 h in a medium supplemented with 10% FBS, as previously described [2]. The non-coding control siRNAs were purchased from Thermo Fisher Scientific (Waltham, MA, USA). The sequences of siRNAs used in this study are listed below (Table 1).

### 2.5. Western Blotting

The proteins were extracted from the cultured cells or mouse tissues as described previously [1,24]. Briefly, 10–50 μg of protein was resolved by SDS-PAGE and transferred to the PVDF membrane. Primary antibodies against INPP4B (Cell signaling, #8450), EZH2 (Cell signaling, #5246), AR (Millipore, #06–680), PTEN (Cell Signaling, #9188), H3K27me3 (Active Motifs, #39157), H3K9me2 (Millipore, #07-441), SMAD4 (Santa Cruz, #7966), p53 (NeoMarker, #MS-187-P1), pAkt S473 (Cell signaling, #4051), total Akt (Cell signaling, #4691), and β-tubulin (Millipore Sigma, #05-661) were used, and the signal was captured by ImageQuant LAS 500 and analyzed by ImageQuant TL (GE Healthcare, Marlborough, MA, USA).

### 2.6. Gene Expression Analysis

RNA isolation and cDNA preparation were described previously [24]. Briefly, RNA was isolated using Tri reagent (Molecular Research Center, Cincinnati, OH, USA) from either the cells or mouse-dissected prostates. cDNA was prepared using a Verso cDNA Synthesis Kit (Thermo Fisher Scientific) as the manufacturer recommended. Real-time PCR was performed using a Roche 480 LightCycler (Roche, Basel, Switzerland). The primers and probes (Appendix A) used in this study are shown in the table below (Table 2). For the cell lines, three or more independent experiments were performed, and at least three biological replicates were applied for each experiment.

### 2.7. Single-Cell RNA-seq Analysis

The scRNA-seq datasets previously reported by Hirz et al. [30] and Henry et al. [31] were retrieved from the Gene Expression Omnibus (GEO). The raw count matrix and cell annotations were downloaded from GSE143791 and GSE181294. Conos [32] was used to process and integrate multiple scRNA-seq datasets. The cells were normalized by the total counts over all the genes followed by log scaling and regressing over the total counts per cell. The expression values of EZH2, INPP4B, and PTEN were obtained for coexpression analysis. EZH2-positive cells from GSE143791 and GSE181294 were extracted and stratified based on the presence of INPP4B, PTEN, and AR.

### 2.8. Animal Husbandry

All the procedures described in this study were approved by the FIU Institutional Animal Care and Use Committee. The animals were housed in an AAALAC-certified facility at FIU. The generation of *Inpp4b^−/−^* FVB mice was described previously [24,33]. Two-month-old male mice were euthanized and their prostates were dissected prior to the RNA and protein extraction or formaldehyde fixation and paraffin embedding.

### 2.9. Immunohistochemistry

The mouse prostates were fixed in 4% formaldehyde (Electron Microscopy Sciences, Hatfield, PA), washed in an ethanol gradient, and embedded in paraffin. The antigen was retrieved by heating in a 0.01 M sodium citrate buffer (pH 6.0). Primary antibodies for EZH2 (Cell signaling, #5246) were used at 1:300 dilution, and sections were counterstained with hematoxylin (EMD Millipore). All the images were acquired using an AxioCam camera and were processed by AxioVision LE software v4.8 (Zeiss).

### 2.10. Statistical Analysis

For the correlation analysis, the expression levels of EZH2 and INPP4B were extracted from the TCGA database [34,35] and from gene sets GSE29079, GSE32448, GSE141551, GSE70770, GSE74367, and were computed using Pearson correlation analysis. The correlation coefficient r and *p* values were calculated using Prism 9. For the bar graph, the data are presented as mean ± SEM. Student t-tests were used for comparing 2 groups using Prism 9. A *p* value < 0.05 was considered statistically significant. For each experiment, at least three biological replicates were performed.

## 3. Results

### 3.1. INPP4B Depletion Reduces EZH2 Expression in Prostate Cancer

To test whether loss of the tumor suppressor INPP4B elicits compensatory mechanisms, we used a reverse phase protein array (PRRA). The levels of 144 proteins were compared in the control and INPP4B knockdown LNCaP cells; the levels of 103 proteins were significantly altered with the loss of INPP4B (Appendix A). Consistent with the INPP4B tumor suppressor function, the tumorigenic markers MAPK, SRC, and SNAI2 were elevated (Figure 1a) [36,37], and the tumor suppressor proteins RB1, BRCA1, and CHEK2 [38,39,40] were decreased in the cells treated with siRNA targeting *INPP4B*.

In contrast, levels of oncogenic EZH2 significantly decreased following *INPP4B* knockdown. In validation, we observed a decrease in *EZH2* mRNA (Figure 1b) and protein levels (Figure 1c) in the LNCaP cells 48 h after *INPP4B* knockdown. Since LNCaP cells do not express PTEN, we used the prostate cancer cell line, VcaP, to compare the effects of INPP4B and PTEN loss on EZH2 protein levels. *INPP4B* knockdown in VCaP cells led to a decrease in EZH2 mRNA and protein levels (Figure 1d,e). Importantly, PTEN knockdown in VCaP cells did not change EZH2 expression (Figure 1d,e).

Using two independent EZH2 signatures derived from EZH2 knockdown GSE39452 and treatment with EZH2 inhibitor EPZ-6438 GSE107779 of LNCaP cells [20,29], we analyzed the INPP4B-regulated genes in the LNCaP (GSE111725) to determine whether INPP4B regulates EZH2-dependent gene expression. The GSEA analysis revealed a highly significant enrichment of both EZH2 transcriptional signatures among the INPP4B-regulated genes (Figure 1f).

### 3.2. Expression of INPP4B Positively Correlates with EZH2 Levels in Human Prostate Epithelium and Primary Tumors

To evaluate the possibility that EZH2 expression is regulated by INPP4B, we tested whether their expression overlaps in the human prostate. The single-cell sequencing data (GSE29079 [31] and GSE32448 [41]) show that *EZH2* and *INPP4B* transcript levels positively correlate in the human prostate epithelium (Figure 2a). An analysis of healthy human prostate single-cell transcriptomics (GSE120716) revealed that EZH2 is expressed in all types of prostate epithelial cells, with the highest level in the luminal and basal subtypes. A portion of *EZH2*-positive prostate epithelial cells expressed *AR* and either *INPP4B* or *PTEN*, but rarely both (Figure 2b). Analysis of an independent single-cell RNA-seq dataset (GSE181294) showed that a similar portion of *EZH2*-positive epithelial, immune, and stromal cells express *INPP4B* and *PTEN* (Appendix A).

Since EZH2 acts as an oncogene in prostate cancer [14,20], we correlated *INPP4B* and *EZH2* expression in indolent and aggressive human prostate cancers. Evaluation of the gene expression data in 499 prostate cancer patients in the TCGA dataset [34,35] showed that *EZH2* expression positively correlates with *INPP4B* (r = 0.1368, *p* = 0.0022), while the correlation is negative with *PTEN* expression (Spearman r = −0.1718, *p* = 0.0001) (Figure 2c,d). Consistent with a hypothesis of compensatory downregulation of *EZH2* in early prostate tumorigenesis, the positive correlation between *INPP4B* and *EZH2* is only evident in the primary prostate tumors (Figure 2e). In advanced metastatic tumors, transcript levels of *INPP4B* and *EZH2* do not correlate (Figure 2f,g).

### 3.3. Androgen Signaling Induces EZH2 in Prostate Cancer Cells

Androgen receptor signaling is the major stimulus for prostate cancer initiation and progression to CRPC [4,42]. Therefore, castration therapies, such as bicalutamide and enzalutamide treatments, are the standard of care for patients with disseminated prostate cancer. Previous reports on mutual regulation of AR and EZH2 are inconsistent (Appendix A). We used an androgen-dependent LNCaP cell line that expresses INPP4B to determine the effect of castration therapies on *EZH2* expression. Early passage LNCaP cells were treated with the AR inhibitors bicalutamide (two days) or enzalutamide (five days). In parallel with a reduction in INPP4B protein expression, EZH2 expression was also reduced by both the enzalutamide and bicalutamide (Figure 3a,b). Androgen deprivation also reduced EZH2 protein and mRNA levels in two independently derived androgen-dependent prostate cancer cell lines: LNCaP (Figure 3c–e) and VCaP (Figure 3d,f).

### 3.4. EZH2 Is Regulated by INPP4B In Vivo in Mouse Prostate

We previously reported that *Inpp4b^−/−^* males develop prostatic inflammation and dysregulation of AR signaling [24,25]. To investigate whether INPP4B regulates EZH2 levels, we compared EZH2 protein levels in prostates of WT and *Inpp4b^−/−^* males by Western blot analysis and immunohistochemistry. Mouse prostates consist of three distinct lobes: the anterior prostate (AP), dorsolateral prostate (DLP), and ventral prostate (VP) [43]. The EZH2 was more highly expressed in the VP of WT males (Figure 4a). In *Inpp4b^−/−^* males, VP EZH2 levels were reduced compared to those of the WT (Figure 4a,b). The immunohistochemical analysis confirmed that the EZH2 protein level is decreased in the secretory prostate epithelium of *Inpp4b^−/−^* males (Figure 4c). In contrast, both benign prostates and prostate tumors in the *Pten^LoxP/LoxP^*, *PB4-Cre* males expressed significantly higher levels of *Ezh2* than the prostates of the WT males (GSE76822, GSE56469, and GSE98493) [7,44,45] (Figure 4d).

EZH2 is a histone methyl transferase that elevates levels of H3K27me3 and H3K9me2 [16,17,18]. We tested whether the downregulation of EZH2 in *Inpp4b^−/−^* prostates decreases the levels of EZH2 methylation targets. Consistent with the pattern of EZH2 expression (Figure 4a), the highest levels of H3K27 tri-methylation and H3K9 di-methylation were observed in the ventral prostate lobe of the wild-type males (Figure 4e). The levels of H3K27me3 and H3K9me2 were significantly decreased in the ventral prostates of the *Inpp4b^−/−^* males compared to those of the WT males (Figure 4e).

### 3.5. INPP4B Loss Leads to a Compensatory Increase in TP53 Protein Expression

The loss of PTEN in the mouse prostate epithelium increases expression of *Trp53* and *Pml* [7,46] and protein levels of SMAD4 [8]. In independently generated gene expression data sets comparing the prostate transcriptomes of WT and *Pten^−/−^* males, both *Trp53* and *Pml* were expressed at higher levels (Figure 5a,b).

In contrast to the loss of *Pten*, we did not observe an increase in SMAD4 protein (Figure 5c), *Pml* (Figure 5d), and *Trp53* (Figure 5e) expression in *Inpp4b^−/−^* males compared to the WT. We and others have shown that the loss of INPP4B leads to activation of Akt, which subsequently activates the mTORC1 complex [2,47,48]. Consistent with previous reports of mTORC1-dependent upregulation of TP53 translation [23], we observed a concomitant increase in prostatic pAkt and p53 protein levels in *Inpp4b^−/−^* males (Figure 5f).

## 4. Discussion

A variety of genomic alterations including mutations, DNA copy number changes, rearrangements, and gene fusions occur as prostate cancer progresses to metastatic disease [4,49,50]. The frequency of these alterations correlates with the shorter time to relapse and with the development of metastases [50]. The INPP4B and PTEN tumor suppressors are lost in advanced disease in 47% and 42% of cancers, respectively [4]. Despite their frequent loss in advanced metastatic prostate cancer, their deletions alone do not produce aggressive prostate cancers in mouse models [5,25,51,52]. While INPP4B and PTEN have similar enzymatic substrates (PIP2 and PIP3, respectively) they regulate overlapping and distinct signaling pathways in the prostate epithelium [1,5,24,51]. Both reduce the phosphorylation of Akt, but only INPP4B also downregulates the activity of PKC [24,25]. *Inpp4b^−/−^* mice fed regular chow do not develop PIN [25]. Prostatic expression of *Pml* and protein levels of SMAD4 (Figure 4) were unchanged, suggesting that, unlike in *Pten^−/−^* males, they do not play compensatory anti-tumorigenic response in that model.

EZH2 is an oncogene in prostate cancer, and its expression increases with the progression to metastatic disease [20]. We showed that *INPP4B* knockdown causes a decrease in EZH2 levels in two independent prostate cancer cell lines, LNCaP and VCaP (Figure 1). We previously showed that *INPP4B* is a direct AR target in LNCaP and VCaP cells [2]. Androgen deprivation or treatment with AR inhibitors causes a reduction in INPP4B mRNA and protein with a subsequent decline in EZH2 levels (Figure 3a–d). A lack of EZH2 downregulation following short-term androgen deprivation or AR antagonist treatments [20,53] suggests that androgen ablation may regulate EZH2 levels indirectly. Its oncogenic potential is dependent on both the lysine methyltransferase activity and the ability to activate AR. In mice, we observed a decrease in EZH2 levels in the ventral prostates of *Inpp4b^−/−^* males (Figure 4). As expected, the reduction in EZH2 resulted in decreased levels of H3K9me2 and H2K27me3 in *Inpp4b^−/−^* males (Figure 4). In our previous report, we demonstrated that *Inpp4b* knockout alters AR signaling in mouse prostates [24]. Consistent with decreased EZH2 levels, expression of the AR and EZH2 co-targeted genes, *Msmb* and *Nkx3.1* [54,55], was significantly decreased in the prostates of *Inpp4b^−/−^* males [24]. This suggests that EZH2 epigenetic and coactivator functions are both affected by INPP4B loss.

In Figure 1, we showed that INPP4B regulates both mRNA and protein levels of EZH2, suggesting transcriptional regulation. Previous studies have shown the regulation of EZH2 by transcription factors, miRNAs, and post-translational modifications [56]. Transcriptional factors like Myc, E2F, EWS-FLI1, SOX4, and NF-KB have been previously shown to regulate EZH2 expression. Additionally, BRD4 has been shown to regulate EZH2 expression in bladder cancer [57]. We compared the gene expression of these transcription factors using an unbiased microarray (GSE111725) generated previously [24]. We found that INPP4B knockdown significantly decreased BRD4, SOX4, and Myc levels in LNCaP cells, which may contribute to the loss of the INPP4B-mediated decrease in EZH2 transcription. Therefore, we hypothesized that the activation of Akt and compound decline in these transcription factors contribute significantly to changes in EZH2 levels and activity.

Similar to prostates of *Pten^LoxP/LoxP^*; *PB4-Cre* males, there is an increase in pAkt and p53 protein levels in prostates of *Inpp4b* knockout animals. It has been previously shown that Akt-dependent activation of the mTORC1 complex leads to increased translation of p53. The p53-dependent cellular senescence has been shown to restrict tumorigenesis in the prostates of *Pten^LoxP/LoxP^*; *PB4-Cre* mice [11]. In prostate cancer, sustained activation of Akt causes p53-dependent senescence, providing selective pressure for the loss of p53 function in PTEN-null tumors. In vivo, INPP4B loss was shown to increase PI(3,4)P2 [47] and PI(3,4,5)P3 [52], both of which bind to the Akt pleckstrin homology domain and activate this kinase. Loss of INPP4B leads to an accumulation of these phosphatidyl inositols and overactivation of Akt [2,25,28]. Activation of the Akt pathway likely contributes to the increase in p53 proteins in *Inpp4b^−/−^* males.

In human prostate samples, we found a highly significant positive correlation between *INPP4B* and *EZH2* in two independent datasets using prostate epithelial cells and in primary tumors. The relatively low r value was likely a contribution of two factors: *EZH2* expression is regulated by multiple pathways, and not all *EZH2* positive cells express *INPP4B* (Figure 2). While there is a significant body of evidence that *EZH2* is downregulated by *PTEN* [55,58], the analysis of single-cell transcriptomic datasets suggests that only a portion of *EZH2*-positive cells express *PTEN*. We observed a comparable level of *INPP4B* expression in *EZH2*-positive cells. Consistent with the opposite modes of regulation, less than 2% of *EZH2*-positive cells express both *INPP4B* and *PTEN*, despite the fact that over 20% of *INPP4B*-positive cells also express *PTEN* (Appendix A).

## 5. Conclusions

In summary, we show that the loss of INPP4B triggers overlapping and distinct compensatory changes in the mouse prostate. INPP4B downregulates p53 levels in mouse prostates and is required for optimal expression of EZH2. In men, there is a positive correlation between the expression of *INPP4B* and *EZH2* in primary prostate cancers. In the mouse prostate, *Inpp4b*, *Trp53,* and *Ezh2* are co-expressed in the epithelial cell. In *Inpp4b^−/−^* mice, decreased EZH2 levels alter the expression of EZH2 and AR co-regulated genes and reduce methylation of the H3 histones.

## Figures and Tables

**Figure 1 cancers-15-05418-f001:**
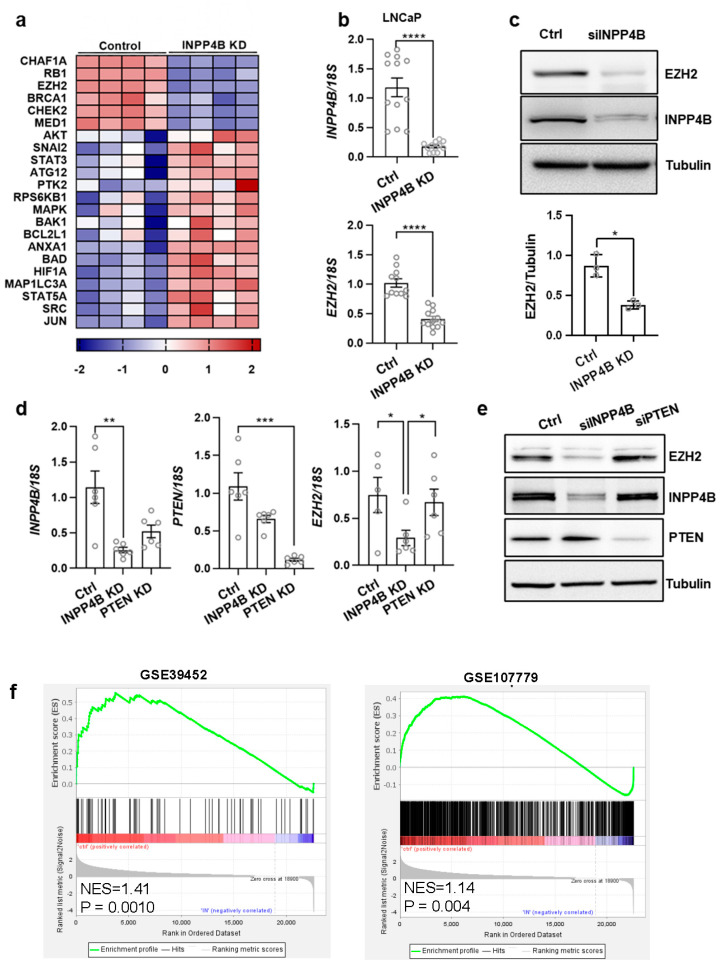
INPP4B expression is positively correlated with EZH2 expression. (**a**) LNCaP cells were transfected with the control or INPP4B siRNA for 48 h. Protein and RNA were extracted and used in a reverse-phase protein array. The heatmap shows the proteins that were significantly altered by INPP4B knockdown. (**b**) RNAs from cells transfected with the noncoding control and INPP4B-specific siRNAs were analyzed for expression of *EZH2* and *INPP4B* using *18S* as the control. (**c**) Cellular lysates from the cells transfected in parallel with (**b**) were assayed for INPP4B, EZH2, and tubulin by Western blotting. The quantification of EZH2 normalized to tubulin is shown below (*N* = 3). (**d**) VCaP cells were transfected with control, INPP4B, or PTEN-specific siRNA for 48 h. RNA and protein were isolated for RT-qPCR and Western blotting. The RNA was analyzed for expression of *INPP4B*, *PTEN*, and *EZH2* using *18S* as the control. (**e**) Protein extracts from cells transfected in parallel with (**d**) were assayed for EZH2, INPP4B, PTEN, and tubulin levels by Western blotting. (**f**) The GSEA of INPP4B-regulated genes using EZH2 transcriptional signatures (Appendix A) in LNCaP cells (GSE39452 and GSE107779). The uncropped blots are in Appendix A. (* *p* < 0.05; ** *p* < 0.01; *** *P* < 0.001; **** *p* < 0.0001).

**Figure 2 cancers-15-05418-f002:**
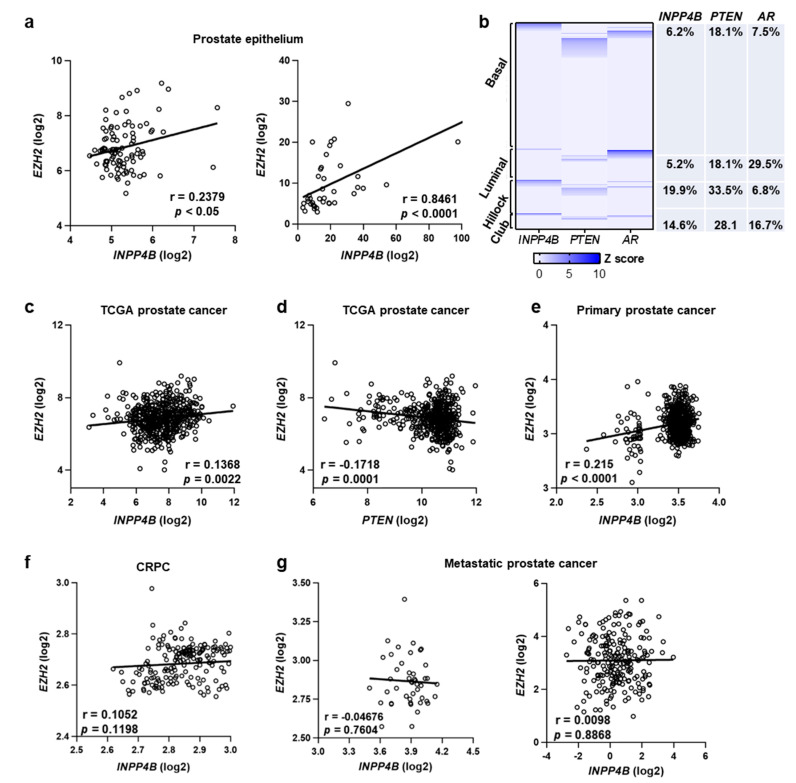
Correlation of EZH2 and INPP4B expression in mice and humans. (**a**) The positive correlation between *INPP4B* and *EZH2* in prostatic epithelium. The expression data were acquired from GSE29079 (*N* = 95) and GSE32448 (*N* = 40). (**b**) Heatmap showing expression of *INPP4B*, *PTEN*, and *AR* in *EZH2*-positive human epithelial cells (left). The expression of *INPP4B*, *PTEN*, and *AR* in *EZH2*-positive cells was normalized using the z-score. The table displays the percentage of EZH2-positive cells that express the indicated genes in each subtype. (**c**) Positive correlation between the expression of INPP4B and EZH2 in 499 human prostate cancer patients (cBioportal, TCGA, 2015). (**d**) Negative correlation between PTEN and EZH2 in the same prostate cancer patients (cBioportal, TCGA, 2015). (**e**–**g**) Positive correlation between *INPP4B* and *EZH2* in primary prostate cancer. (**e**) No correlation was detected between *INPP4B* and *EZH2* in CRPC (**f**) and metastatic prostate cancer datasets (**g**). Data were acquired from GSE141551 (*N* = 548), GSE70770 (*N* = 220), GSE74367 (*N* = 45), and TCGA databases (*N* = 212).

**Figure 3 cancers-15-05418-f003:**
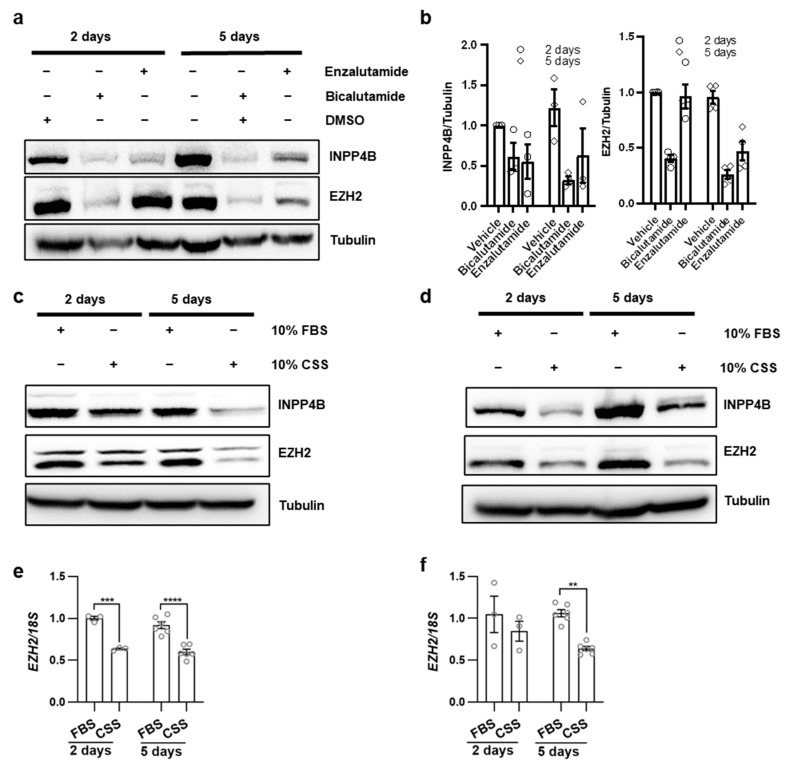
Anti-androgen treatments reduce EZH2 levels in prostate cancer cell lines. (**a**,**b**) LNCaP cells were plated in a complete growth medium. Cells were treated with DMSO, 10^−6^ M bicalutamide, or 10^−5^ M enzalutamide for 2 and 5 days as indicated. Cell lysates were assayed for INPP4B, EZH2, and tubulin using Western blotting. (**b**) Quantification of INPP4B and EZH2 normalized to tubulin from (**a**). (**c**,**d**) LNCaP (**c**) or VCaP (**d**) cells were plated in a medium supplemented with either 10% FBS or 10% CSS for 2 and 5 days. Proteins were extracted and assayed for INPP4B, EZH2, and tubulin using Western blotting. (**e**,**f**) Comparison of EZH2 expression in LNCaP (**e**) and VCaP (**f**) cells incubated in 10% FBS or 10% CSS for 2 and 5 days. The uncropped blots are in Appendix A. (** *p* < 0.01; *** *p* < 0.001; **** *p* < 0.0001).

**Figure 4 cancers-15-05418-f004:**
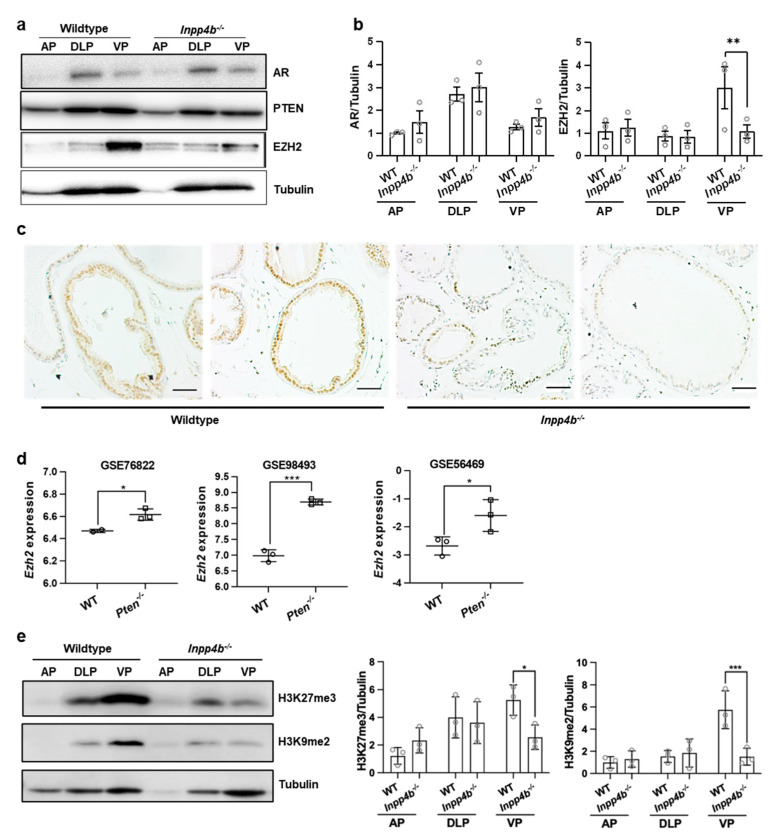
The loss of INPP4B reduced EZH2 levels and activity in the mouse ventral prostate. (**a**) Protein was extracted from the AP, DLP, and VP from WT or *Inpp4b^−/−^* males and analyzed for AR, PTEN, EZH2, and tubulin levels by Western blotting. (**b**) Quantification of the protein levels in (**a**) (*N* = 3 for EZH2 and *N* = 3 for AR) (**c**) Prostates from two-month-old WT or *Inpp4b^−/−^* males were stained with EZH2 antibodies and counterstained with hematoxylin. Scale bars represent 20 µm. (**d**) Expression of *Ezh2* was compared in prostates of WT mice and mice with prostate-specific deletions of *Pten*. Data acquired from GSE76822, GSE56469, and GSE98493. (**e**) Protein was extracted from the AP, DLP, and VP from WT or *Inpp4b^−/−^* males and analyzed for H3K27me3, H3K9me2, and tubulin levels by Western blotting. Quantification of H3K27me3 and H3K9me2 are shown on the right of the Western blot image (*N* = 3). The uncropped blots are in Appendix A. (* *p* < 0.05; ** *p* < 0.01; *** *p* < 0.001).

**Figure 5 cancers-15-05418-f005:**
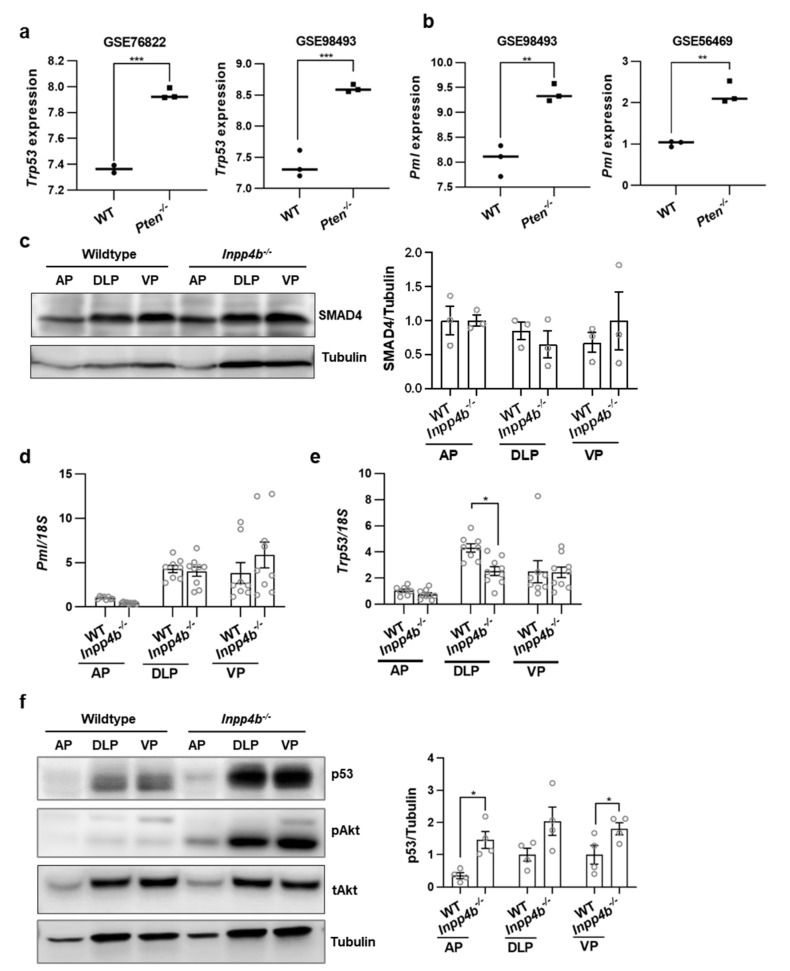
TRP53 levels are elevated in prostates of *Inpp4b^−/−^* mice. (**a**,**b**) Expression of (**a**) *Trp53* and (**b**) *Pml* were compared in prostates of WT and *Pten^LoxP/LoxP^; PB4-Cre* males (GSE76822, GSE56469, and GSE 98493). (**c**) Protein was extracted from the AP, DLP, and VP from WT or *Inpp4b^−/−^* males and analyzed for SMAD4 and tubulin levels by Western blotting. Quantification of SMAD4 is shown on the right (*N* = 3). (**d**,**e**) RNA was isolated for RT-qPCR from the AP, DLP, and VP from WT or *Inpp4b^−/−^* males and analyzed for (**d**) *Pml* and (**e**) *Trp53* using *18S* as the control. (**f**) Protein was extracted from the AP, DLP, and VP from WT or *Inpp4b^−/−^* males and analyzed for p53, pAkt, total Akt, and tubulin levels by Western blotting. Quantification of p53 is shown on the right (*N* = 4). The uncropped blots are in Appendix A. (* *p* < 0.05; ** *p* < 0.01; *** *p* < 0.001).

**Table 1 cancers-15-05418-t001:** siRNA sequences.

siRNA	Sense	Antisense
INPP4B	GAGCCUGAACUGCAUUAUU	AAUAAUGCAGUUCAGGCUC
INPP4B	CGAUGUCAGUGACACUUGA	UCAAGUGUCACUGACAUCG
PTEN	CCAUUACAAGAUAUACAAU	AUUGUAUAUCUUGUAAUGG
PTEN	AAACAUUAUUGCUAUGGGA	UCCCAUAGCAAUAAUGUUU

**Table 2 cancers-15-05418-t002:** Primers and probes for qRT-PCR.

Gene name	Forward primer	Reverse primer	Probe
INPP4B (human)	tgtctgatgctgacgctaaga	ccacaaaccaatccagcaa	41
PTEN (human)	ggggaagtaaggaccagaga	tccagatgattctttaacaggtagc	48
EZH2 (human)	tgtggatactcctccaaggaa	gaggagccgtcctttttca	35
*Ezh2* (mouse)	gaataacagtagcagacccagca	gcttctctgtcactgtctgtatcc	109
*Pml* (mouse)	cccaacctgtggctatggta	ccttgcattgaaaaggcatac	1
*Trp53* (mouse)	gcaactatggcttccacctg	ttattgaggggaggagagtacg	4
18S	gcaattattccccatgaacg	gggacttaatcaacgcaagc	48

## Data Availability

The sources of data acquired from public repositories are described in the manuscript. The R script used to analyze sc-RNA data is available upon request from the corresponding author/Shenglin Mei.

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
