# Peer review of "Regulation of EZH2 Expression by INPP4B in Normal Prostate and Primary Prostate Cancer"

_cancers, 2023, doi:10.3390/cancers15225418_

Round 1
Reviewer 1 Report (Previous Reviewer 1)
Comments and Suggestions for Authors
The authors has addressed most of my questions
Reviewer 2 Report (Previous Reviewer 3)
Comments and Suggestions for Authors
Previous concerns are mostly addressed and this time it has been composed more delicately. No further issues.
This manuscript is a resubmission of an earlier submission. The following is a list of the peer review reports and author responses from that submission.
Round 1
Reviewer 1 Report
Comments and Suggestions for Authors
Zhang et al described in this manuscript a few interesting observations that INPP4B regulates EZH2 expression in early-stage prostate cancer. Although EZH2 is an important regulator in prostate cancer progression, this manuscript lacks mechanistic studies of how INPP4B (a lipid phosphatase) modulates EZH2 expression at mRNA and protein levels. Some other issues may improve this manuscript as well.
1. Can authors include the data on changes in INPP4B in prostate cancer patients in Figure 1, including the types of changes, frequency, and corresponding survival curves? Although this manuscript mentioned the presence of a significant loss of INPP4B in prostate cancer patients, more direct data will better illustrate the physiological significance of this gene's function. 2. In Figure 1D, the level of INPP4B decreases when PTEN is knocked down, and when INPP4B is knocked down, the level of PTEN also decreases. Usually, when the expression of one of the functionally similar proteins is reduced, the expression of the other should increase to compensate for the functional defect, Please have a reasonable explanation. 3. Please include the results of AR expression level changes in the ARSI and CSS treatments shown in Figure 3 as a control. This will provide better evidence for the regulation of EZH2 by AR signaling. 4. In Figures 3A/C/D, were the results for day 2 and day 5 obtained from consecutive processing, or were they from two separate experiments? If they were obtained from consecutive processing, why did the INPP4B levels in Figure 3D show a rebound on the 5th day after CCS treatment? 5. Does the inhibition of the AR signaling pathway also affect the expression of PTEN? Will the loss of INPP4B affect the response of cells to AR signaling pathway inhibitors? 6. Why are TP53, PML, and SMAD4 validated in different ways in Figure 5? (At the protein level and mRNA level). Why is there a discrepancy between the expression of TP53 protein and mRNA in Figure 5E/F? 7. EZH2 is a protein that plays a role mainly in prostate cancer progression and advanced prostate cancer, and it is also mentioned in the paper that INPP4B deficiency accumulates in patients with advanced prostate cancer. So, does INPP4B also have a regulatory role for EZH2 in advanced prostate cancer (e.g., CRPC)?
It is unfortunate that the specific mechanism by which INPP4B regulates EZH2 expression is not tested and if provided, will much improve this manuscript.
Comments on the Quality of English LanguageOverall, the language of the essay is clear and understandable. It does not require much revision and can be embellished where appropriate to improve overall readability.
Reviewer 2 Report
Comments and Suggestions for Authors
Brief Summary: The aim of the study by Zhang et al., was to investigate molecular compensatory mechanisms that may be activated to suppress prostatic tumorigenesis after Inositol polyphosphate-4-phospatase type II B (INPP4B) loss. Based on their previous studies (PMID:30228349), the authors utilized a mouse model harboring a germline Inpp4b loss (Inpp4b -/-), analyses of human prostate cancer cell lines and molecular datasets to show that INPP4B loss may be involved in the regulation of distinct compensatory molecular pathways contributing to suppression of prostate cancer progression. They show that Inpp4b downregulates p53 and Ezh2 levels in mouse prostates, while INPP4B expressions correlates with EZH2 levels in primary prostate cancer. Although not very compelling, the authors also provide some data on the comparison between the tumorigenic activities of Pten and Inpp4b loss, suggesting that either regulates distinct compensatory pathways that limit progression to more aggressive disease. Overall, this is a well-written study that presents potentially interesting data on molecular mechanisms that may be hindering prostate cancer progression. However, the translational relevance of these findings is unclear, and the manuscript requires revisions to be considered for publication.
Strengths of the study:
· Description of new compensatory molecular pathways that may hinder prostate cancer progression.
· Use of mouse and human datasets for the analyses.
Weaknesses of the study:
· Unclear translational relevance to prostate cancer, especially since their previous study (PMID:30228349) described in detail similar molecular pathways related to INPP4B loss.
· The comparison between INPP4B and PTEN loss activities for prostate cancer is interesting but requires further clarification and investigation.
Comments:
Simple summary/Abstract: The authors summarize their study nicely.
Introduction: This section presents background information on prostate cancer, INPP4B, PTEN and EZH2.
· However, the authors should provide more references indicating the “strong” association of both INPP4B and PTEN for prostate cancer tumorigenesis. Does their claim come from patient dataset analyses (e.g. TCGA PRAD)? [minor]
Materials and Methods: The experimental procedures and study design are comprehensively explained.
Results and Figures: Overall, the results and figures are described and presented well. Comments:
· In Figure 2, the spearman correlation panels could use additional titles to help the reader distinguish the comparisons/datasets. [minor]
· In Figure 3, panels a, c and d need re-formatting. [minor]
· In Figure 5, panels c and f the tubulin loading controls for the DLP appear not to be equal to the rest. Consider re-running these experiments to provide more robust protein expression data. [minor]
· The authors used 2-month-old Inpp4b -/- for their gene expression and phenotypic analyses, although the strongest diseased phenotype in these mice is observed at 12-months (PMID:30228349). They should clarify and justify their choice, especially when comparing these data with gene expression datasets derived from Ptenflox/flox prostate of older mice. The comparisons made could be unequal and biologically irrelevant. Relevant to this point, the histological images of the 2-month-old Inpp4b -/- prostates in Figure 4c, do not seem different compared to wildtype. [major]
· Two different gene expression datasets of conditional Ptenflox/flox or germline Pten-/- mice were used in the study. Since the Inpp4b -/- mice are also germline, it seems more reasonable to utilize the germline Pten-/- datasets for the majority of the comparisons and not conditional Ptenflox/flox. The authors should explain their choice and/or perform additional analyses. [major]
· Rb1 expression appears to be down in prostate cancer cells with loss of INPP4B (Figure 1a). It will be interesting to determine the status of Rb1 in the prostates of the Inpp4b -/- mice and discuss why, if downregulated in their mouse model, it is not enough to drive more aggressive prostate cancer phenotypes. [minor]
· In the analyses of the TCGA dataset, the authors should provide additional information on the clinical correlates regarding INPP4B expression (e.g. Gleason score and BCR-free survival) to strengthen the translational relevance of the study. [major]
· The authors should discuss why EZH2 protein expression in the LNCaP cells is reduced in androgen deprivation conditions (Figure 3c), whereas in other published studies (PMID: 23239736) the opposite is observed. [minor]
Discussion and Conclusions: The authors present and adequately discuss their findings in the most part.
· This section would benefit from some further discussion on the points mentioned above [major].
· The percentages of INPP4B and PTEN loss in advanced disease mentioned here are not supported by a citation. The authors should provide the appropriate citation. [minor]
Reviewer 3 Report
Comments and Suggestions for Authors
Zhang et al. have explored the regulation of EZH2 expression by INPP4B in normal prostate and primary prostate cancer. They find that knockdown of INPP4B, leads to a decrease in EZH2 expression in human prostate cancer cell lines. The authors also observe a positive correlation between INPP4B and EZH2 expression in normal human prostates and prostate tumors. This work provides a regulatory pathway for prostate cancer progression and potentially a therapeutic approach. However, there are a few questions and comments the authors should address.
Major
1) There is only one figure in the supplement material, and the rest are missing.
2) A major concern is the significance and clinical impact of this work. The knockdown of INPP4B increased the expression of several oncogenic proteins and reduced the expression of tumor suppressor proteins. However, the knockdown was also associated with reduced expression of EZH2. So, what does it mean for tumors presenting a loss of INPP4B (a suppressor), which also leads to a reduced expression of EZH2 (an oncogenic protein)? Does the harm of losing INPP4B and related tumor suppressors overtake the benefit of losing EZH2? Or vice versa? This is the part that is missing to demonstrate the significance of this work, and the authors should elicit that comprehensively. What does it mean for us to understand and develop therapeutics targeting this pathway?
3) Additionally, how does the tumor behave when losing INPP4B and with the downstream pathways altered? Does it become more malignant or more benign? This could easily be addressed with an inoculation experiment using the cell lines treated with the siRNA.
4) For all the western blots using samples from the anterior prostate (AP), the loading was extremely low. This will affect the reading of the relative expression even with normalization. Please adjust the sample concentration or volume. Also, N=2 for EZH2 western blot does not have enough statistical power, please increase the N.
Minor
1) There are misalignments for the labels and their corresponding western blot lanes throughout the figures.
2) Line 263, based on Figure 3 and its legend, the cell line was treated with bicalutamide or enzalutamide for both 2 days and 5 days, but the text in this line implies the cells were treated with bicalutamide for 2 days and enzalutamide for 5 days.
3) Please label the panels in Figure 2 with their categories, e.g., prostate epithelium, prostate tumor, primary tumor, and metastatic tumor, in the title of each chart.
